# Population-Based Estimates of the Age-Specific Cumulative Risk of Breast Cancer for Pathogenic Variants in *CHEK2*: Findings from the Australian Breast Cancer Family Registry

**DOI:** 10.3390/cancers13061378

**Published:** 2021-03-18

**Authors:** Tú Nguyen-Dumont, James G. Dowty, Jason A. Steen, Anne-Laure Renault, Fleur Hammet, Maryam Mahmoodi, Derrick Theys, Amanda Rewse, Helen Tsimiklis, Ingrid M. Winship, Graham G. Giles, Roger L. Milne, John L. Hopper, Melissa C. Southey

**Affiliations:** 1Precision Medicine, School of Clinical Sciences at Monash Health, Monash University, Melbourne, Victoria 3168, Australia; tu.nguyen-dumont@monash.edu (T.N.-D.); jason.steen@monash.edu (J.A.S.); anne-laure.renault@monash.edu (A.-L.R.); Fleur.Hammet@monash.edu (F.H.); maryam.mahmoodi@monash.edu (M.M.); derrick.theys@monash.edu (D.T.); amanda.rewse@monash.edu (A.R.); Helen.Tsimiklis@monash.edu (H.T.); Graham.Giles@cancervic.org.au (G.G.G.); Roger.Milne@cancervic.org.au (R.L.M.); 2Department of Clinical Pathology, Melbourne Medical School, University of Melbourne, Melbourne, Victoria 3010, Australia; 3Centre for Epidemiology and Biostatistics, Melbourne School of Population and Global Health, University of Melbourne, Melbourne, Victoria 3010, Australia; jdowty@unimelb.edu.au (J.G.D.); j.hopper@unimelb.edu.au (J.L.H.); 4Royal Melbourne Hospital, Parkville, Melbourne, Victoria 3050, Australia; Ingrid.Winship@mh.org.au; 5Department of Medicine, The University of Melbourne, Melbourne, Victoria 3010, Australia; 6Cancer Epidemiology Division, Cancer Council Victoria, Melbourne, Victoria 3004, Australia

**Keywords:** *CHEK2*, breast cancer, predisposition, genetic risk factors, age-specific cumulative risk, penetrance

## Abstract

**Simple Summary:**

It is well established that women who carry pathogenic *CHEK2* variants have about a 3-fold increased risk of developing breast cancer. *CHEK2* is now commonly included in genetic tests for breast cancer predisposition and increasingly used to inform the clinical management of women who are identified to carry pathogenic variants. Important information for counselling these women includes knowing how breast cancer risk, due to having a pathogenic variant in *CHEK2*, changes over a woman’s lifetime. This information is currently not well established. By conducting a population-based case-control-family study of pathogenic *CHEK2* variants we aimed to provide this information and estimated the penetrance (age-specific cumulative risk) of breast cancer to be 18% (95% CI 11–30%) to age 60 years and 33% (95% CI 21–48%) to age 80 years. These findings provide new and important information for the clinical management of breast cancer risk for women carrying pathogenic variants in *CHEK2*.

**Abstract:**

Case-control studies of breast cancer have consistently shown that pathogenic variants in *CHEK2* are associated with about a 3-fold increased risk of breast cancer. Information about the recurrent protein-truncating variant *CHEK2* c.1100delC dominates this estimate. There have been no formal estimates of age-specific cumulative risk of breast cancer for all *CHEK2* pathogenic (including likely pathogenic) variants combined. We conducted a population-based case-control-family study of pathogenic *CHEK2* variants (26 families, 1071 relatives) and estimated the age-specific cumulative risk of breast cancer using segregation analysis. The estimated hazard ratio for carriers of pathogenic *CHEK2* variants (combined) was 4.9 (95% CI 2.5–9.5) relative to non-carriers. The HR for carriers of the *CHEK2* c.1100delC variant was estimated to be 3.5 (95% CI 1.02–11.6) and the HR for carriers of all other *CHEK2* variants combined was estimated to be 5.7 (95% CI 2.5–12.9). The age-specific cumulative risk of breast cancer was estimated to be 18% (95% CI 11–30%) and 33% (95% CI 21–48%) to age 60 and 80 years, respectively. These findings provide important information for the clinical management of breast cancer risk for women carrying pathogenic variants in *CHEK2*.

## 1. Introduction

Cell-cycle checkpoint kinase 2 (*CHEK2*) is an established breast cancer predisposition gene whose protein plays an important role in cell-cycle checkpoint regulation and DNA damage repair [1,2]. *CHEK2* c.1100delC is the most common protein-truncating variant in European populations and has been the focus of work that has estimated the magnitude of breast cancer risk associated with *CHEK2* pathogenic variants [3,4,5,6]. A meta-analysis of breast cancer risk associated with *CHEK2* c.1100delC estimated an odds ratio (OR) of 2.7 (95% CI, 2.1–3.4) for unselected breast cancer cases and OR 4.8 (95% CI, 3.3–7.2) for women with a family history of breast cancer [6]. *CHEK2* c.1100delC carriers have shorter breast cancer-specific survival compared with non-carriers [7,8,9,10,11] and have a higher risk of contralateral breast cancer [8,9,12,13]. Schmidt et al. used the resources of the Breast Cancer Association Consortium (BCAC) including 44,777 women with breast cancer and 42,997 unaffected women from 33 studies and estimated the odds ratio (OR) for invasive breast cancer as 2.30 (95% Confidence Interval, 1.90–2.69) which was predominantly a risk for estrogen receptor (ER) positive disease OR 2.55 (95% CI, 2.10–3.10) [14]. The estimated cumulative risks for the development of ER-positive disease for carriers of *CHEK2* c.1100delC was 20% to age 80 years [14].

Although *CHEK2* c.1100delC has been investigated in population-based studies of younger women with breast cancer, few have used methodologies that are demonstrated to provide unbiased estimates [3,6,8]. Much less information has been reported for other protein-truncating variants and missense variants in *CHEK2*. Le Calvez-Kelm et al. [15] performed mutation screening in *CHEK2* in a population-based series of 1303 women with breast cancer and 1109 unaffected controls. This work found that there were *CHEK2* rare variants (in addition to *CHEK2* c.1100delC) associated with increased breast cancer risk and a substantial proportion of these were missense variants [16]. Six of the rare *CHEK2* missense variants identified by Le Calvez-Kelm et al. were incorporated in a custom Illumina Infinium array (iCOGS) as part of a multiconsortia collaboration [17,18]. Genotyping of 42,671 invasive breast cancer cases and 42,164 controls from BCAC found evidence that both *CHEK2* c.349A>G and c.538C>T were associated with increased breast cancer risk; OR 2.26 (95% CI, 1.29–3.95) and OR 1.33 (95% CI, 1.05–1.67) respectively [19].

*CHEK2* is now included in routine gene panel tests for breast cancer susceptibility which has substantially increased the need to more precisely understand the breast cancer risks associated with rare genetic variants in this gene. As described above, previous work has predominantly estimated breast cancer risk in terms of average relative risks and has been either exclusive to, or dominated by, risks associated with *CHEK2* c.1100delC. For the purposes of genetic counseling, in the era of gene panel testing, age-specific cumulative risks (penetrance) are more useful. Often, absolute risk estimates are derived from relative risk estimates, which are likely to be biased due to oversampling by many studies of cases with a family history [20].

To address this knowledge gap, we conducted a genetic screen of *CHEK2* in an Australian population-based case-control-family study of breast cancer. This study is focused on disease at an early age and participants were unselected for family history. We estimated the prevalence and penetrance of *CHEK2* pathogenic variants in this study sample and discuss the capacity of these findings to further inform risk management strategies for families carrying these pathogenic variants.

## 2. Results

### 2.1. Pathogenic CHEK2 Variants Identified in the Australian Breast Cancer Family Registry

Targeted-sequencing was successfully performed on the germline DNA of 1476/1480 (99.7%) case-probands and 861/864 (99.7%) control-probands. A pathogenic (including likely pathogenic) variant in *CHEK2* was identified in 20/1476 (1.4%) case-probands and 7/861 (0.8%) control-probands. There were two carriers of a stop-gain variant, 17 carriers of a frameshifting variant, two carriers of a consensus splice site variant and six carriers of a missense variant (Table 1). Of the 17 frameshifting variant carriers, 12 carried *CHEK2* c.1100delC (nine case-probands and three control-probands).

A total of 74 family members, from 16 of the 27 families, had germline DNA available for predictive testing. Of these 74 family members, 30 were found to carry the pathogenic *CHEK2* variant identified in the proband (Table 1).

### 2.2. Risk Estimates

One of the 27 *CHEK2* pathogenic carrier families (the case-proband family with splice donor variant NM_007194.3:c.319+2T>A) also carried a pathogenic variant in *BRCA1*, so this family was excluded from the risk analyses. There were, therefore, 26 eligible families, which included 1071 relatives of the proband, 72 of which had DNA for testing (described above). The risk estimates were effectively based on the 1071 relatives of the remaining 26 probands. These 1071 relatives included 43 female breast cancer cases and one male breast cancer case, though the male case did not affect our estimates because male hazard ratio (HRs) were fixed to be 1. In addition to the 44 relatives with breast cancer, a number of relatives had cancers of other anatomical sites: 16 prostate, 16 bone marrow, 11 intestine, 10 stomach, 9 skin, 8 colorectum, 7 lung, 6 brain and 49 at all other sites combined (with 5 or fewer at any individual site). The relatives included 28 known carriers and 44 known non-carriers, though ungenotyped people also contributed to our estimates via their phenotypes and their relationships to genotyped people.

The estimated breast cancer HR for carriers of pathogenic *CHEK2* variants was 4.9 (95% CI 2.5–9.5; *p* < 0.0001), meaning that carriers are estimated to have breast cancer incidences that are 4.9 times those for non-carriers. There was no evidence that the HR depended on age (*p* = 1). The HR for carriers of the *CHEK2* c.1100delC variant was estimated to be 3.5 (95% CI 1.02–11.6) and the HR for carriers of all other *CHEK2* variants combined was estimated to be 5.7 (95% CI 2.5–12.9), though our data were consistent with carriers of the c.1100delC variant having the same risks as carriers of the other *CHEK2* variants (*p* = 0.5).

Based on the above HR estimate for all pathogenic *CHEK2* variants combined, cumulative risks for these carriers to various ages were calculated (Table 2, Figure 1). Carriers have an estimated 2.6% and 33% probability of developing breast cancer by the age of 40 years and 80 years, respectively.

Sensitivity analyses showed that our main results were robust to changes in the assumed allele frequency within a plausible range. The HR (95% CI) for all pathogenic *CHEK2* variants combined was estimated to be 4.89 (2.5–9.6), 4.85 (2.5–9.5) and 4.79 (2.4–9.4) when the combined allele frequency was taken to be 0.001, 0.005 and 0.01, respectively.

## 3. Discussion

This important new information supports gene panel testing and clinical management, including genetic counselling, for women who carry pathogenic variants in *CHEK2*.

In gene panel sequencing for breast cancer predisposition, *CHEK2* is consistently found to have a high number of pathogenic variants, usually only surpassed by the number of pathogenic variants identified in *BRCA1* and *BRCA2* in most settings not involving selection by breast cancer subtype [21,22]. The prevalence of pathogenic *CHEK2* variants in this population-based study (1.4% of case probands) is consistent with reports from other studies in which panel testing was applied.

Case-control studies have estimated a 3-4-fold increased risk of breast cancer associated with pathogenic *CHEK2* variants [21,23,24,25,26]. This has firmly consolidated *CHEK2* as *a bona fide* breast cancer predisposition gene but information suitable for estimating age-specific cumulative risk (penetrance) has been lacking. Our population-based case-control-family study provides such information and estimated penetrance of *CHEK2* pathogenic variants to be 26% (95% CI, 16–40%) and 33% (95% CI, 21–40) to 70 and 80 years respectively. There is no directly comparable data (generated with similar study design and methodology) from previous literature. The most relevant is the estimate of cumulative risk of breast cancer for carriers of *CHEK2* c.1100delc (only) reported by Schmidt et al (2016) who estimated 22% by age 80 [14]. This estimate falls at the lower end of the 95% confidence interval of our estimate at age 80 (95% CI 21–48) but our analysis is not restricted to *CHEK2* c.1100delC.

Other breast cancer susceptibility genes that have equivalent information include *BRCA2* (penetrance to age 70, 45% (95% CI, 31–56%) [27], *PALB2* (penetrance to age 70, 44% (95% CI, 37–52%) [28] and *BRCA1* (penetrance to age 70, 65% (95% CI, 44–78%) [27]. The methodology applied in this report (population-based family studies and a mixed model, which incorporated an unmeasured polygene in addition to the major gene), has previously demonstrated the capacity to provide robust penetrance estimates [29,30,31]. Following the example of other initiatives, we should now pool family data with other studies to refine penetrance estimates for breast and other cancers (e.g., [27,28]). These findings are timely as the inclusion of *CHEK2* on standard cancer predisposition gene panel tests means that many clinical genetics services are already using data related to *CHEK2* pathogenic variants to advise women and their families about breast cancer risk management. Appropriate risk management strategies are predicated on robust risk estimates; these include surveillance, but also irreversible clinical decisions such as risk reducing mastectomy.

There are limitations to our study. Genetic testing was limited to the coding regions of *BRCA1, BRCA2* and *CHEK2* and the amplicon-based approaches used in this study only enable the detection of single nucleotide substitutions and small insertion and deletions. The probands participating in the ABCFR resources have previously been systematically screened for large genomic rearrangements in *BRCA1* and *BRCA2* [32,33], but not in *CHEK2* and thus unidentified carriers of large rearrangement events in *CHEK2* may exist in the study resources. Our analyses were limited to variants that were classified as pathogenic in ClinVar. An expert panel oversees *BRCA1* and *BRCA2* variant classification reported in ClinVar but the equivalent ClinVar expert panel (and rules) for classification of variants in *CHEK2* is still under development. Furthermore, *CHEK2* missense variants have not been investigated as extensively by functional assays compared to missense variants *BRCA1* and *BRCA2*. With our capability for clinical classification of *CHEK2* variants still emerging, many studies have focused on *CHEK2* c.1100delC. The rarity of other variants greatly limit our ability to provide precise estimates of risk on a variant by variant basis. The estimated HRs are instead provided for pathogenic variants as a group. Lastly, the parametric survival analysis built into our segregation analysis models assumed the age at breast cancer diagnosis was independent of the age of death from other causes (i.e., non-informative censoring), which could very slightly bias our penetrance estimates at older ages.

Further international collaboration is required to confirm and refine the breast cancer risk estimates and further understand breast cancer risk modifying factors. Important factors include family history, which still remains an important additional clinical consideration for families that carry pathogenic variants in *PALB2* [28] and the role of the polygenic risk score that has, similar to *BRCA1* and *BRCA2* pathogenic variant carriers, recently been demonstrated to modify the risk of breast cancer for *CHEK2* pathogenic variant carriers [34].

It is also important to further understand the risk of other cancers (in addition to breast cancer) that are associated with pathogenic variants in *CHEK2* that include cancers in the Li-Fraumeni syndrome spectrum [35] and prostate cancer [36] and the associated penetrance of *CHEK2* pathogenic variants for these cancers. Furthermore, it is likely that the presence of pathogenic variants in the *CHEK2* gene will impact management options for those in whom detection coincides with the diagnosis of breast cancer, in consideration of targeted chemotherapy and the safety of radiotherapy.

## 4. Materials and Methods

### 4.1. Study Subjects

The Australian Breast Cancer Family Registry (ABCFR) is a population-based, case-control-family study of breast cancer, carried out in Australia (Melbourne and Sydney) as part of the international Breast Cancer Family Registry (BCFR). Case probands were over-sampled for those with early-onset breast cancer.

The BCFR is an infrastructure for cooperative multinational, interdisciplinary and translational studies of the genetic epidemiology of breast cancer supported by the National Cancer Institute (USA) since 1995 [37,38]. The broad aims of the BCFR are to expedite the translation of genetic epidemiological research findings to affected and at-risk populations. All adult women living in the metropolitan areas of Melbourne and Sydney who were diagnosed with an incident, histologically-confirmed, first primary cancer of the breast were invited to participate in the ABCFR. Recruitment was irrespective of their family history of breast cancer.

Cases were identified via the Victorian and New South Wales cancer registries, to which notification of all cancer diagnoses is a legislative requirement. Cancers in relatives were verified by cancer registry reports, medical records or death certificates. Control subjects were adult women living in the metropolitan areas of Melbourne and Sydney who were selected from the electoral roll (by use of proportional random sampling based on the expected age distribution of the case probands), and approached with a letter similar to that used for case probands [39].

All participants provided written informed consent for participation in the study. This study was approved by The University of Melbourne Human Research Ethics Committee (approval number: 1441420).

### 4.2. Targeted-Sequencing in Proband Subjects

Blood-derived germline DNA from 1480 case probands and 864 control probands were screened by targeted-sequencing of the coding regions and proximal intron–exon junctions of BRCA1 (NM_007294.3), BRCA2 (NM_000059.3) and CHEK2 (NM_007194.3). Libraries were prepared using one of two amplicon-based methodologies, Hi-Plex2 [40] or Halo-PlexHS (Agilent, Santa Clara, CA, USA) and sequenced on the NextSeq550 (Illumina, San Diego, CA, USA) or HiSeq3000 (Illumina), respectively. The BED files corresponding to the targeted regions are available upon request.

### 4.3. Sequencing Data Processing and Variant Selection

Paired-end reads were aligned to the human reference genome GRCh37 using BWA-mem 0.7.17. [41]. HaloPlex-derived alignment files were processed using the Agilent Genomics NextGen Toolkit to remove adapter sequences and mark molecular barcodes. Hi-Plex alignment files did not undergo any further processing. Target coverage was calculated using bedtools [42]. Samples with ≥80% target bases covered at ≥50X sequencing depth were considered successfully sequenced. Single nucleotide variants (SNVs) and short insertion/deletions (indels) were called using VarDict [43]. Further analysis was restricted to variants in the coding regions and consensus splice sites of the genes included on the panel that had a read depth ≥50X and 30X for samples produced using the Hi-Plex and HaloPlex methodologies, respectively, and a variant allele frequency (VAF) ≥ 0.2. Variant annotation was performed using VarSeq VSClinical v2.2 (Golden Helix Inc., Bozeman, MT, USA).

This study focused on rare pathogenic or predicted protein-truncating variants. Rare variants were defined as those identified in the Exome Aggregation Consortium (ExAC) with a minor allele frequency (MAF) ≤ 0.01 in the non-Finnish European population (NFE-non TCGA). Genetic variants in CHEK2 were considered pathogenic if they were annotated as “Pathogenic” or “Likely Pathogenic” in ClinVar (accessed 30 July 2020). Predicted protein-truncating variants in CHEK2 that were absent from ClinVar were included in this analysis, except if they were located in the last coding exon. For BRCA1 and BRCA2 variants, pathogenicity was strictly determined using the ENIGMA expert panel classification available from ClinVar.

### 4.4. Targeted-Sequencing in Family Members

DNA was extracted from blood samples provided by family members of case-probands and control-probands that were identified to carry pathogenic variants in CHEK2. Targeted-sequencing of CHEK2, BRCA1 and BRCA2 using the Hi-Plex panel and variant annotation and classification were conducted as described in Section 4.2 and 4.3, respectively.

### 4.5. Statistical Methods

Segregation analysis [27,44,45] was used to estimate hazard ratios (HRs) for all carriers of pathogenic *CHEK2* variants, where the HR is defined to be the age-specific breast cancer incidences for female carriers divided by those for non-carriers. Segregation analysis is a classical method for analysing family data [27,44,45] that allows all family members to be included in the analysis, whether genotyped or not, using a rigorous approach that accounts for all possible combinations of joint genotypes within each family that are consistent with the known genotypes. All segregation analysis models were fitted by the method of maximum likelihood, as implemented in the statistical package MENDEL version 3.2 [46]. All estimates were appropriately adjusted for the population-based ascertainment of families using retrospective likelihood [47], in which each pedigree’s data were conditioned on the proband’s genotype, breast cancer status and age of onset. See our previous report [48] for the mathematical details of an approach that is similar to the one used here.

The age at breast cancer diagnosis was modelled as a random variable whose hazard is the relevant HR multiplied by the incidence of breast cancer for non-carriers. Non-carrier incidences were chosen so that the average incidence for carriers and non-carriers (weighted by the carrier frequency) was the age-specific population incidence rates for Australia in the period 1998–2002, as obtained from *Cancer Incidence in Five Continents* [49]. Participants were considered to be at risk of breast cancer from birth until the earliest of breast cancer diagnosis, death or last follow-up. The allele frequency of all pathogenic *CHEK2* variants combined was taken to be 0.005, which is approximately half the carrier frequency in controls [50,51,52], though sensitivity analyses were conducted to assess the impact of the allele frequency on our main results. An age dependence for the HR was investigated by modelling the HR as continuous, piece-wise linear functions of age that are locally constant before age 40 years and after age 60 years, and linear in between. HRs for males were fixed to be 1 in all analyses.

Major gene models, which attribute all familial aggregation of disease to the gene being studied, can often give biased estimates of risk [31]. Therefore, a mixed model, which incorporates an unmeasured polygene in addition to the major gene (*CHEK2*) [44,53], was used in all modified segregation analyses to model any residual familial aggregation of breast cancer risk (see the detailed methods in a previous report [48]). The polygenic part of this mixed model, which models the combined effect of a large number of genes that individually have small, additive effects on the log HR, was implemented as a hypergeometric polygenic model with four loci [44,53]. Under this model, the polygene is approximately normally distributed (on the log HR scale) and is correlated within families, with correlation coefficients equal to the kinship coefficients [45]. The mean of the polygene was chosen so that the average HR was 1. The standard deviation of the polygene was fixed to be 1.17741, which corresponds under Pharoah’s formula to a breast cancer familial relative risk of 2 [54].

The age-specific cumulative risk (penetrance) to age t years was calculated from the estimated HR as
$$1-\exp\left(-\overset{t}{\underset{0}{\int }}\lambda \left(s\right)ds\right),$$
where λ(s) is the relevant incidence rate averaged over the polygene, i.e., for carriers, λ(s) is the HR at age s multiplied by the non-carrier incidence at age s. Confidence intervals (CIs) for these cumulative risks were calculated by evaluating the above expression for the cumulative risk at the terminals of the CI of the HR (for 1-parameter models) or by using a parametric bootstrap (for multi-parameter models). For the parametric bootstrap: a sample of 5000 draws were taken from the multivariate normal distribution that the maximum likelihood estimates would be expected to follow under asymptotic likelihood theory; for each age, a corresponding sample of 5000 cumulative risks to that age was calculated using the formula above; then, the 95% CI for the cumulative risks to a given age were taken to be the 2.5 th and 97.5 th percentiles of this sample.

No ages at diagnosis were missing for the breast cancer cases, though ages at death or last follow-up were missing for 219 (21.2%) of the 1032 controls. The missing ages for these people (who were mostly second-degree relatives of the proband) were set to 0, which censored them at birth and effectively removed them from the analysis. Non-missing ages were truncated at 80 years.

All *p*-values were 2-sided, and all *p*-values for the segregation analyses were based on the likelihood ratio test. A *p*-value threshold of 0.05 was used to define statistical significance.

## 5. Conclusions

We estimated the age-specific cumulative risk of breast cancer for carriers of pathogenic variants in *CHEK2* in a population-based case-control-family study. As gene panel testing is now becoming routine and implementation of population screening is increasingly discussed, these risk estimates are highly relevant and urgently needed for the clinical management of the women and families found to carry a pathogenic variant in *CHEK2*.

## Figures and Tables

**Figure 1 cancers-13-01378-f001:**
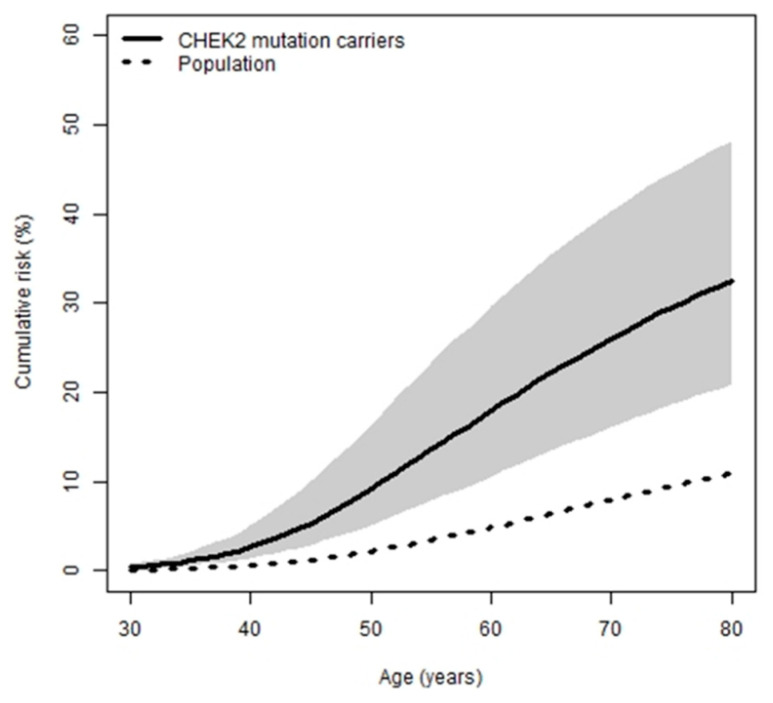
Average age-specific cumulative risk (penetrance) of breast cancer, for Australian women (dashed line) and for female carriers of pathogenic and likely pathogenic *CHEK2* variants combined (solid line), with confidence intervals for carriers (grey region).

**Table 1 cancers-13-01378-t001:** Pathogenic and likely pathogenic ^a^
*CHEK2* variants identified by targeted-sequencing in the case and control probands participating in the Australian Breast Cancer Family Registry.

	Variant Type	HGVS.c ^b^	HGVS.p ^b^	Number of Relatives Who Are Carriers/Tested/Total	Number of Relatives with Breast Cancer Who Are Carriers/Tested/Total
Case proband	Nonsense	NM_007194.3:c.1528C>T	NP_009125.1:p.Gln510Ter	3/5/34	1/1/3
		NM_007194.3:c.823G>T	NP_009125.1:p.Glu275Ter	1/1/97	0/0/5
	Frameshift	NM_007194.3:c.1263delT	NP_009125.1:p.Ser422Valfs *15	0/0/40	0/0/1
		NM_007194.3:c.1263delT	NP_009125.1:p.Ser422Valfs *15	3/5/78	2/2/3
		NM_007194.3:c.1100delC	NP_009125.1:p.Thr367Metfs *15	2/6/46	1/2/4
		NM_007194.3:c.1100delC	NP_009125.1:p.Thr367Metfs *15	2/3/20	0/0/0
		NM_007194.3:c.1100delC	NP_009125.1:p.Thr367Metfs *15	0/5/16	0/0/1
		NM_007194.3:c.1100delC	NP_009125.1:p.Thr367Metfs *15	4/17/67	2/5/10
		NM_007194.3:c.1100delC	NP_009125.1:p.Thr367Metfs *15	2/3/41	0/0/0
		NM_007194.3:c.1100delC	NP_009125.1:p.Thr367Metfs *15	1/4/75	0/1/1
		NM_007194.3:c.1100delC	NP_009125.1:p.Thr367Metfs *15	1/1/19	0/0/0
		NM_007194.3:c.1100delC	NP_009125.1:p.Thr367Metfs *15	2/2/33	0/0/0
		NM_007194.3:c.1100delC	NP_009125.1:p.Thr367Metfs *15	0/0/19	0/0/0
		NM_007194.3:c.920dupG	NP_009125.1:p.Glu308Argfs *4	1/1/17	0/0/0
		NM_007194.3:c.405delA	NP_009125.1:p.Lys135Asnfs *26	0/0/47	0/0/2
	Splice donor	NM_007194.3:c.444+1G>A		0/0/18	0/0/0
		NM_007194.3:c.319+2T>A ^c^		2/2/23	0/0/0
	Missense	NM_007194.3:c.349A>G	NP_009125.1:p.Arg117Gly	3/4/32	2/2/2
		NM_007194.3:c.349A>G	NP_009125.1:p.Arg117Gly	2/12/190	1/1/6
		NM_007194.3:c.349A>G	NP_009125.1:p.Arg117Gly	1/3/24	0/0/1
Control proband	Frameshift	NM_007194.3:c.1100delC	NP_009125.1:p.Thr367Metfs *15	0/0/41	0/0/2
		NM_007194.3:c.1100delC	NP_009125.1:p.Thr367Metfs *15	0/0/19	0/0/0
		NM_007194.3:c.1100delC	NP_009125.1:p.Thr367Metfs *15	0/0/32	0/0/0
		NM_007194.3:c.591delA	NP_009125.1:p.Val198Phefs *7	0/0/23	0/0/1
	Missense	NM_007194.3:c.349A>G	NP_009125.1:p.Arg117Gly	0/0/21	0/0/1
		NM_007194.3:c.349A>G	NP_009125.1:p.Arg117Gly	0/0/13	0/0/0
		NM_007194.3:c.349A>G	NP_009125.1:p.Arg117Gly	0/0/9	0/0/1

^a^ According to ClinVar accessed July 2020. ^b^ Variant nomenclature according to the Human Genome Variation Society (HGVS), HGVS.c for coding DNA and HGVS.p for protein variants, based on transcript sequence NM_007194.3, +1 as A of ATG start codon. ^c^ Proband also found to carry a *BRCA1* nonsense variant (NM_007294.3:c.1840A>T).

**Table 2 cancers-13-01378-t002:** Average age-specific cumulative risk (penetrance) of breast cancer to various ages, for female, Australian carriers of pathogenic and likely pathogenic *CHEK2* variants (combined).

Age (Years)	Cumulative Risk (%) to Each Given Age, with 95% Confidence Intervals in Parentheses
30	0.4 (0.2–0.7)
40	2.6 (1.4–5.1)
50	9.1 (5.1–16)
60	18 (11–30)
70	26 (16–40)
80	33 (21–48)

## Data Availability

Data is available upon request to the corresponding author.

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
