# Peer review of "Population-Based Estimates of the Age-Specific Cumulative Risk of Breast Cancer for Pathogenic Variants in CHEK2: Findings from the Australian Breast Cancer Family Registry"

_cancers, 2021, doi:10.3390/cancers13061378_

Round 1

Reviewer 1 Report

Thank you for addressing the comments, these have improved and clarified the man

Minor revisions:

 - At line 39 in the abstract, can you please remove the p-value, it is not needed here.

- In the abstract, can you please add the number of families and individuals included in the study.

Author Response

We have made these edits in our manuscript.

Reviewer 2 Report

The authors have thoroughly addressed my prior suggestions.

Author Response

Thank you for your helpful suggestions.

This manuscript is a resubmission of an earlier submission. The following is a list of the peer review reports and author responses from that submission.

Round 1

Reviewer 1 Report

The authors have done a commendable job addressing an important research gap estimating the age-specific cumulative risk of breast cancer for women with pathogenic CHEK2 variants using a population based case-control family study. A few minor comments that will help frame this manuscript:

  1. On line 65 the authors state “few unbiased age-specific risk estimates have been reported” yet do not describe at all how their estimates are unbiased and how their study is different than previous studies. Including this in the discussion would be helpful.
  2. When estimating the age-specific cumulative risk of breast cancer – did you take into account competing risk? If so, how did you do this and competing risk from which other outcomes (death, or other cancers that are associated with CHEK2)? Please explain.
  3. The discussion really lacked much discussion about the limitations of the study – please include this.
  4. On line 162, you refer to the polygenic risk as the “so-called polygenic risk”. What do you mean by so-called? Do you mean to say the polygenic risk score? Please elaborate or just say polygenic risk score.

Author Response

  1. On line 65 the authors state “few unbiased age-specific risk estimates have been reported” yet do not describe at all how their estimates are unbiased and how their study is different than previous studies. Including this in the discussion would be helpful.

To address this important comment we have edited the text at lines 65-67 to read “Although CHEK2 c.1100delC has been investigated in population-based studies of younger women with breast cancer, few have used methodologies that are demonstrated to provide unbiased estimates [3,6,8]. This text precedes lines 82-84 “Often absolute risks estimates are derived from relative risk estimates, which are likely to be biased due to oversampling by many studies of cases with a family history [20]” There is further reference to this issue in the methods section at lines 306-310 “Major gene models, which attribute all familial aggregation of disease to the gene being studied, can often give biased estimates of risk [31]. Therefore, a mixed model, which incorporates an unmeasured polygene in addition to the major gene (CHEK2) [44,53], was used in all modified segregation analyses to model any residual familial aggregation of breast cancer risk (see the detailed methods in a previous report [48]).”

We have added the following text to the discussion line 176 “The methodology applied in this report (population-based family studies and a mixed model, which incorporated an unmeasured polygene in addition to the major gene) has previously demonstrated capacity to provide robust penetrance estimates [29-31].”as suggested by the reviewer.

  1. When estimating the age-specific cumulative risk of breast cancer – did you take into account competing risk? If so, how did you do this and competing risk from which other outcomes (death, or other cancers that are associated with CHEK2)? Please explain.

We censored the study subjects at the competing event of death from any cause (and also at last follow-up) but we did not explicitly model competing risks, since this would have been difficult given our limited number of families and the current lack of knowledge about the effect of CHEK2 variants on the risk of death or other cancers.  Censoring at death is very common in penetrance studies, but it assumes that the age at breast cancer diagnosis is independent of the age of death from other causes (i.e. non-informative censoring).  This could lead to small biases at older ages (small compared to the imprecision of the estimates, which is a much bigger issue), so we have added a comment about this to the discussion.  Note that no study is unbiased in a strict sense, e.g. explicitly modelling the censoring process would introduce small biases if the censoring process is mis-specified in small ways, which it always is.  

  1. The discussion really lacked much discussion about the limitations of the study – please include this.

Thank you for the suggestion. The following paragraph discussing the limitations of the study has been added:

There are limitations to our study. Genetic testing was limited to the coding regions of BRCA1, BRCA2 and CHEK2 and the amplicon-based approaches used in this study only enable the detection of single nucleotide substitutions and small insertion and deletions. The ABCFR resources have previously been systematically screened for large genomic rearrangements in BRCA1 and BRCA2 (but not CHEK2)[Smith et al., 2007 and Smith et al 2011] and thus unidentified carriers of large rearrangement events in CHEK2 may exist in the study resources. Our analyses were limited to variants that were classified as pathogenic in ClinVar. An expert panel oversees BRCA1 and BRCA2 variant classification reported in ClinVar but the equivalent ClinVar expert panel (and rules) for classification of variants in CHEK2 is still under development.  Furthermore, CHEK2 missense variants have not been investigated as extensively by functional assays compared to missense variants BRCA1 and BRCA2. With our capability for clinical classification of CHEK2 variants still emerging, many studies have focused on CHEK2 c.1100delC.  The rarity of other variants greatly limit our ability to provide precise estimates of risk on a variant by variant basis. The estimated HRs are instead provided for pathogenic variants as a group. Lastly, the parametric survival analysis built in to our segregation analysis models assumed the age at breast cancer diagnosis was independent of the age of death from other causes (i.e. non-informative censoring), which could very slightly bias our penetrance estimates at older ages.”

  1. On line 162, you refer to the polygenic risk as the “so-called polygenic risk”. What do you mean by so-called? Do you mean to say the polygenic risk score? Please elaborate or just say polygenic risk score.

Thank you.  We have changed the text to say “polygenic risk score” as suggested.

Author Response

  1. Please be more specific on how it was verified that the control-probands did not have history of cancers. Also, was it verified that controls had received recent negative breast cancer screening results?

The ABCFR selected control probands from the electoral rolls for participants living in Melbourne and Sydney. Adult registration for voting is compulsory in Australia. Notification of a cancer diagnosis is also compulsory under legislation and is considered complete. The ABCFR has also conducted regular active follow-up of families, through phone interviews, mailed questionnaires, clinic visits and linkage to cancer registries. This is described in detail in Hopper et al 1999. We have now added this reference.

  1. It would be useful to present a comparison of the characteristic of the case and control families, including age, degree of relatedness of included relatives, sex distribution and paternal/maternal family distribution to assess how similar those samples are.

The ABCFR resources have been described in detail elsewhere including McCredie et al 1998 and John et al 2004 (now both cited). However, it is not clear if the reviewer is specifically asking about the families that carry P/LP variants in CHEK2 – if this is the case some of this information is contained in the revised table 1 but further information could be provided in an additional table upon request.

  1. I did not find information on family history of other cancers in the case-probands and control-probands families to check for differences (especially colorectal, prostate and kidney cancers which have been associated with CHEK2). The authors did not discuss whether information was collected or if it would impact their estimates.

Our report estimates the penetrance of breast cancer (only) and is insufficiently sized to consider cancers outside of breast cancer. We have edited the text in the discussion as follows;

Following the example of other initiatives, we should now pool family data with other studies to refine penetrance estimates for breast and other cancers (e.g., [27,28]).

In response to this comment and those from other reviewers we have also included some detail about the other cancers in relatives in these families in the results section at lines 128-131, as follows: 

“In addition to the 44 relatives with breast cancer, a number of relatives had cancers of other anatomical sites:  16 prostate, 16 bone marrow, 11 intestine, 10 stomach, 9 skin, 8 colorectum, 7 lung, 6 brain and 49 at all other sites combined (with 5 or fewer at any individual site).”

  1. There is no description of whether MLPA was used to detect large rearrangements in BRCA1/2 or whether other methods were used. If large rearrangements were not assessed, there remains a possibility that some of the probands were carriers of a BRCA1/2 pathogenic variant.

The probands participating in the ABCFR have been screened for large genomic rearrangements in BRCA1 and BRCA2 in previous research (eg., Smith et al., 2011 and Smith et al., 2007).  Large rearrangements in CHEK2 were not assessed in this study, as amplicon-based panel testing does not enable the detection of this type of genetic variant. We thus cannot rule out that some of the participants carry large rearrangements in CHEK2 however, it is highly unlikely that CHEK2pathogenic variant carriers identified in the panel testing also carry large rearrangements in BRCA1 or BRCA2. This is now acknowledged as a limitation to this study as follows.

The probands participating in the ABCFR have previously been systematically screened for large genomic rearrangements in BRCA1 and BRCA2 (but not CHEK2) and thus unidentified carriers of large rearrangement events in CHEK2 may exist in the study resources.”

  1. The study design does not allow to rule out the possibility that family members might carry a pathogenic variant in another gene (it is my understanding that only CHEK2 variants are screened in relatives).

Family members were tested for CHEK2, BRCA1 and BRCA2. This was described in the Material and Methods and now edited for clarity (gene names have been added). One proband carrier of a pathogenic CHEK2 variant had a relative who was found to also carry a pathogenic BRCA1 variant – this family was excluded from the statistical analyses (lines 110-112).

  1. It is difficult to assess how many relatives were included in the analysis and this could be clarified in the write up. There is mention of 1,071 relatives but CHEK2 carrier status seems to be known for only 28+44 (28 known carriers and 44 non carriers).

After excluding the family of a carrier of a pathogenic BRCA1 variant (lines 110-112), there were 26 families with a proband carrier of a pathogenic CHEK2 variant. In these 26 eligible families, there were 1,071 relatives participating in the ABCFR and DNA was available for testing for 72 of these 1,071 individuals. All of the 1,071 relatives contributed to the analysis, as currently stated (see our response to “What were the criteria to determine an individual is an obligate carrier?”, below). The text has been modified to read:

There were, therefore, 26 eligible families which included 1071 relatives, 72 of which had DNA for testing (described above).” (lines 112-113).

Are the genotypes known for all the relatives listed in table 1?

No, some affected relatives were not able to be genotyped.

How many additional non-genotyped people contributed to the analysis?

1,071participants – 72 participants genotyped = 999 individuals not genotyped.

We have edited table 1 to include the following information;

Number of relatives who are carriers

Number of relatives who were tested

Total number of relatives

Number of relatives with breast cancer who were carriers

Number of relatives with breast cancer who were tested

Total number of relatives with breast cancer

What were the criteria to determine an individual is an obligate carrier? Were the affected relatives assumed to be carriers even in absence of genotype data? If this is the case, it is an assumption that is likely to introduce error as it is not rare to see sporadic cases in the family of carriers of moderate penetrance variants. It should also be specified whether relatives from both paternal and maternal family are included and in what proportion.

We didn’t identify obligate carriers, or make any assumptions about affected relatives being carriers, or impute genotypes in any other way.  Instead, we used a rigorous and sophisticated statistical method for analysing family data with incomplete genotyping, called segregation analysis.  This is a very well-established method of statistical genetics (e.g. see (Lange, 2002)) and it is known to give unbiased penetrance estimates when used as we use it here (Gong et al., 2010).  This method considers all possible combinations of genotypes for the members of each family that are consistent with the known genotypes (there are typically millions of these possible joint genotypes) and the method effectively weights each possible genotype combination by its probability of occurring, based on the phenotypes (ages and affected statuses), family structure, and known genetic laws of inheritance (such as Mendel’s laws).  All members of each family are included in the analysis, so 100% of relatives from the paternal and maternal sides of each family are included.  We have now added some comments about this to the start of the statistical methods.

  1. There is preliminary evidence associating CHEK2 (at least the c.1100del variant) with an increased risk of breast cancer in males, which is not accounted for in this analysis.

Unfortunately, our study wasn’t large enough to estimate a separate breast cancer hazard ratio for males, because there was only 1 male breast cancer in all of our families.  Due to the rarity of male breast cancer, much larger studies than ours would be required to estimate these risks.  Please also see response to question 3 above.

  1. The clinical utility of table 2 would be increased by presenting the data as age range (for example 30-39).

  An age range would be incorrect in Table 2, because this table presents the probability of carriers developing breast cancer by particular ages.  We explain how to interpret this table in the results using the example “carriers have an estimated 2.6% and 33% probability of developing breast cancer by the age of 40 years and 80 years, respectively”.  We’re therefore not sure how to incorporate this suggestion. 

  1. The sample sizes are limited and does not allow to evaluate independently truncating variants from missense variants. While an analysis comparing the c.1100delC variant to others is presented, it is of limited impact given those limitations.

Yes, the sample size does not allow for extensive sub-analyses and or variant specific analyses as explained in the discussion. “The rarity of [other] variants greatly limit our ability to provide precise estimates of risk on a variant by variant basis. The estimated HRs are instead provided for pathogenic variants as a group.”

Discussion:

  1. The cohort is described as having an ‘emphasis on early-onset breast cancer’. Do the authors foresee that this could impact their ascertainment to higher-risk case probands and have an impact on HR estimate?

The phrase “emphasis on early-onset breast cancer” means simply that probands were over-sampled for those with early-onset breast cancer, as now indicated in Study subjects.  We conditioned on the probands’ ages at onset and affected status, which our study’s ascertainment criteria are a function of, so our estimates are not biased by ascertainment, e.g. see (Kraft &Thomas, 2000).   We have edited the text as follows;

“The Australian Breast Cancer Family Registry (ABCFR) is a population-based, case-control-family study of breast cancer (probands were over-sampled for those with early-onset breast cancer), carried out in Australia (Melbourne and Sydney) as part of the international Breast Cancer Family Registry (BCFR).”

  1. There is very limited discussion of the impact of the family history of cancers on the risk estimates, especially cancer types associated with CHEK2 such as colorectal, kidney and prostate. Is the family history of those cancer likely to have an impact on the penetrance of CHEK2? There is one short sentence about it but needs to be further discussed. Is the data available to factor this in the analysis?

 Our report estimates the penetrance of breast cancer (only) and is insufficiently sized to consider cancers outside of breast cancer. We have edited the text in the discussion as follows;

“Following the example of other initiatives, we should now pooled family data with other studies to refine penetrance estimates for breast and other cancers (e.g., [27,28]).”

In response to this and other similar comments from the reviewers we have also included some detail of what other cancers were observed in the families that were found to carry pathogenic CHEK2 variants as follows at lines 128-131;

“In addition to the 44 relatives with breast cancer, a number of relatives had cancers of other anatomical sites:  16 prostate, 16 bone marrow, 11 intestine, 10 stomach, 9 skin, 8 colorectum, 7 lung, 6 brain and 49 at all other sites combined (with 5 or fewer at any individual site).”

  1. I would suggest adding a paragraph to the discussion with the limitations of the study. While the data to address the comments above might not be available, it could be discussed in this paragraph.

Thank you for the suggestion. We have added a paragraph discussing the limitations of our study as follows;

There are limitations to our study. Genetic testing was limited to the coding regions of BRCA1, BRCA2 and CHEK2 and the amplicon-based approaches used in this study only enable the detection of single nucleotide substitutions and small insertion and deletions. The ABCFR resources have previously been systematically screened for large genomic rearrangements in BRCA1 and BRCA2 (but not CHEK2)[ Smith et al., 2007; Smith et al., 2011] and thus unidentified carriers of large rearrangement events in CHEK2 may exist in the study resources. Our analyses were limited to variants that were classified as pathogenic in ClinVar. An expert panel oversees BRCA1 and BRCA2 variant classification reported in ClinVar but the equivalent ClinVar expert panel (and rules) for classification of variants in CHEK2 is still under development.  Furthermore, CHEK2 missense variants have not been investigated as extensively by functional assays compared to missense variants BRCA1 and BRCA2. With our capability for clinical classification of CHEK2 variants still emerging, many studies have focused on CHEK2 c.1100delC.  The rarity of other variants greatly limit our ability to provide precise estimates of risk on a variant by variant basis. The estimated HRs are instead provided for pathogenic variants as a group. Lastly, the parametric survival analysis built in to our segregation analysis models assumed the age at breast cancer diagnosis was independent of the age of death from other causes (i.e. non-informative censoring), which could very slightly bias our penetrance estimates at older ages.”

  1. What are the current clinical guidelines for the management of CHEK2 carriers and how are the presented results likely to impact them?

There is great variation in guidelines for the management of CHEK2 P/LP variant carriers nationally and internationally.  There is also variation in the way these guidelines are applied by individual practitioners. This is in part due to the lack of the type of data presented in this report. This is also in part due to the recent and rapid accumulation of data and the only periodic updating of guidelines. In Australia, risk management guidelines are available for unaffected known or obligate CHEK2 truncating pathogenic variant carriers and those at 50% risk of inheriting such a variant (only). 

These results are likely to advance the implementation of guidelines for carriers of P/LP variants, outside of c.1100delC, as it provides more precise estimates of the penetrance of all P/LP CHEK2 variants (not just c.1100delC).

Reviewer 3 Report

The MS “Population-based estimates of the age-specific cumulative risk of breast cancer for pathogenic variants in CHEK2: findings from the Australian Breast Cancer Family Registry” by Tu Nguyen-Dumont et al, describes a population-based case-control-family study of pathogenic CHEK2 variants (beyond CHEK2 c.1100delC) aiming to provide risk information and estimate the penetrance.

The study is well designed with adequate calculation of estimations.

The need of this data for clinical decisions is well supported in the discussion and conclusion.

It is well known the difficulties to get relatives to be tested, and here (1071 non-tested vs 74 tested, 44 from them resulted non-carriers) it is not an exception. This wide lack of testing was elegantly overcoming in the introduction and discussion.

About half of the references are more than 10 years old. There is reasonable published recent data that could be incorporated in the discussion to upgrade the MS although it should not compromise its publication, this is to be taken as a minor point.

Author Response

About half of the references are more than 10 years old. There is reasonable published recent data that could be incorporated in the discussion to upgrade the MS although it should not compromise its publication, this is to be taken as a minor point.

The authors think that the relevant literature is included in this manuscript after responding to the other three reviewer comments.

Reviewer 4 Report

Dear Authors,

Congratulations on your study, and thank you for giving me the opportunity to review the manuscript. The aim of the study was to estimate prevalence and penetrance of CHEK2 variants in an Australian population-based case-control-Family study of breast cancer. HR for carriers of the CHEK2 c.1100delC variant was found to be 3.5, and for all other CHEK2 variants 5.7. The estimated age-specific cumulative risk of breast cancer was 18% at 60 years and 33% at 80 years. Knowledge of the breast cancer risk associated with CHEK2 c.1100delC and other pathogenic variants in these gene is needed to develop guidelines for follow-up of female carriers. This manuscript adds to the literature adressing this issue.

I have a few comments regarding the manuscript.

Material and Methods:

1.The data from the study is taken from the Australian Breast Cancer Family Registry (ABCFR), which is defined as a "population-based, case-control-Family study of breast cancer With an emphasis on early-onset breast cancer". I think it is relevant to explain what "an emphasis on early- onset breast cancer" means. In another study based on this material (Reference number 29 in the manuscript), breast cancer patients diagnosed at 40 years or younger are included. Is the same age cut off used in this study?  May the Young age of onset of the cases have affected the results in any way?

It is also described that the cases were identified via the Victorian and New South Wales cancer registries. It is not Clear to me what the relationship between the ABCFR and the Victorian and New South Wales cancer registries is. Are women included who are in the registry and who also have consented to inclusion in ABCFR?

2. It is not clearly stated whether  the probands are included in the calculations.

Results:

In table 1, number of relatives and number of relatives found to be carriers are listed. I think it is relevant also to list how many female relatives there are.

Discussion:

  1. The estimated HR for the CHEK2 variant 1100delC is lower than that for other CHEK2 variants combined, but the authors conclude that carriers of the 1100delC variants have similar risk as carriers of other variants. I think this theme could be addressed in the discussion, as knowledge of risk associated with other variants is mentioned as a knowledge gap in the introduction.
  2. In the beginning of the discussion, the authors state that the information from this study supports gene panel testing and clinical management for women who carry pathogenic variants in CHEK2. I think it is relevant for the authors to discuss how their estimates compare to other and previous findings, and whether their findings support a continuation of current guidelines or not.
  3. The sentence “This small study that should be complemented by following the example of others that have pooled family data to refine penetrance estimates for pathogenic variants in breast cancer predisposition genes the ABCFR has previously demonstrated capacity to provide robust penetrance estimates” is a bit long and confusing. Can you clarify?
  4. In the last section, the authors discuss possible risk of other cancers that could be associated with pathogenic variants in CHEK2. This section needs to be referenced. Is it also relevant to list other cancers observed in the included families?

Author Response

Material and Methods:

  1. The data from the study is taken from the Australian Breast Cancer Family Registry (ABCFR), which is defined as a "population-based, case-control-Family study of breast cancer With an emphasis on early-onset breast cancer". I think it is relevant to explain what "an emphasis on early- onset breast cancer" means. In another study based on this material (Reference number 29 in the manuscript), breast cancer patients diagnosed at 40 years or younger are included. Is the same age cut off used in this study?  May the Young age of onset of the cases have affected the results in any way?

The phrase “emphasis on early-onset breast cancer” means simply that probands were over-sampled for those with early-onset breast cancer, as now indicated in Study subjects.  We conditioned on the probands’ ages at onset and affected status, which our study’s ascertainment criteria are a function of, so our estimates are not biased by ascertainment, e.g. see (Kraft &Thomas, 2000). 

We have edited the text as follows to clarify this issue;

The Australian Breast Cancer Family Registry (ABCFR) is a population-based, case-control-family study of breast cancer (probands were over-sampled for those with early-onset breast cancer), carried out in Australia (Melbourne and Sydney) as part of the international Breast Cancer Family Registry (BCFR).

  1. It is also described that the cases were identified via the Victorian and New South Wales cancer registries. It is not Clear to me what the relationship between the ABCFR and the Victorian and New South Wales cancer registries is. Are women included who are in the registry and who also have consented to inclusion in ABCFR?

It is a legislative requirement for all cancers to be reported to the relevant state Cancer Registry in Australia.  Case participants were identified by use of the Victorian and New South Wales Cancer Registries (Victoria and New South Wales are states of Australia). These state registries invited women with breast cancer diagnoses to participate in the ABCFR and those who accepted this invited provided informed consent to participate. Further detail is provided in [33].

  1. It is not clearly stated whether the probands are included in the calculations.

Our risk analyses were conditioned on the probands’ genotypes and phenotypes (ages and affected statuses).  More precisely, our estimates were obtained by maximizing a conditional likelihood, conditional on the probands’ genotypes and phenotypes (our ascertainment criteria).  Conditioning like this is a sophisticated method for preventing ascertainment bias (Gong et al., 2010).  It is not possible to explain the exact effect of conditioning in non-technical terms but, to a good approximation, conditioning is effectively the same as removing the probands from the analysis.  So strictly speaking, the statistical analyses included all probands and all relatives, though the probands were effectively removed from the analysis by conditioning.  

We now say the following in 3.2 Risk estimates the following text and (lines 113-114).

The risk estimates were effectively based on the 1071 relatives of the remaining 26 probands.”

  1. In table 1, number of relatives and number of relatives found to be carriers are listed. I think it is relevant also to list how many female relatives there are.

Unfortunately, there is not enough space in Table 1 to add another column, though the number of female relatives is roughly half the total number of relatives (and similarly for the number of female carriers).  We chose the existing columns because the statistical power of our methods depends strongly on the total number of people and the number of genotyped people in the families, but the power only depends weakly on the number of female carriers.  This is similar to the way that genotype probabilities within a family depend on the family structure and known genotypes, but usually not the sex of the carriers.   

Discussion:

  1. The estimated HR for the CHEK2 variant 1100delC is lower than that for other CHEK2 variants combined, but the authors conclude that carriers of the 1100delC variants have similar risk as carriers of other variants. I think this theme could be addressed in the discussion, as knowledge of risk associated with other variants is mentioned as a knowledge gap in the introduction.

We have edited the discussion to address this issue and similar issues raised by other reviewers.

  1. In the beginning of the discussion, the authors state that the information from this study supports gene panel testing and clinical management for women who carry pathogenic variants in CHEK2. I think it is relevant for the authors to discuss how their estimates compare to other and previous findings, and whether their findings support a continuation of current guidelines or not.

As presented in the manuscript, there no directly comparable data (generated with similar study design and methodology), in previous literature.  The most relevant is the estimate of cumulative risk of breast cancer for carriers of CHEK2pathogenic variants reported by Schmidt et al (2016) [14] that was limited to CHEK2 c.1100delC. The authors modelled the CHEK2 c.1100delC breast cancer risk estimates by age by using the estimates for age from only the cases of a case-control study and estimated a cumulative risk of about 22% by age 80.  This estimate falls at the lower end of the 95% confidence interval of our estimate at age 80 (95% CI 21-48) but our analysis is not restricted to CHEK2 c.1100delC. We have added the following text to address this point;

There is no directly comparable data (generated with similar study design and methodology) from previous literature. The most relevant is the estimate of cumulative risk of breast cancer for carriers of CHEK2 c.1100delc (only) reported by Schmidt et al (2016) [14] who estimated 22% by age 80. This estimate falls at the lower end of the 95% confidence interval of our estimate at age 80 (95% CI 21-48) but our analysis is not restricted to CHEK2 c.1100delC.”

As discussed above, there is considerable variation in guidelines nationally and internationally related to the management of carriers of CHEK2 P/LP variants. 

  1. The sentence “This small study that should be complemented by following the example of others that have pooled family data to refine penetrance estimates for pathogenic variants in breast cancer predisposition genes the ABCFR has previously demonstrated capacity to provide robust penetrance estimates” is a bit long and confusing. Can you clarify?

To address this and other reviewer comments we have edited the text as follows:

The methodology applied in this report (population-based family studies and a mixed model, which incorporated an unmeasured polygene in addition to the major gene), has previously demonstrated capacity to provide robust penetrance estimates [29-31]. Following the example of other initiatives, we should now pool family data with other studies to refine penetrance estimates for breast and other cancers (e.g., [27,28]).”

  1. In the last section, the authors discuss possible risk of other cancers that could be associated with pathogenic variants in CHEK2. This section needs to be referenced.

We have referenced the text in this section. 

Is it also relevant to list other cancers observed in the included families?

Our report estimates the penetrance of breast cancer (only) and is insufficiently sized to consider cancers outside of breast cancer. However, we do understand that the reader may be interested (as were the reviewers), in knowing what other cancers have been observed in these families.  We have included the following text in the results section to provide further information;

“In addition to the 44 relatives with breast cancer, a number of relatives had cancers of other anatomical sites:  16 prostate, 16 bone marrow, 11 intestine, 10 stomach, 9 skin, 8 colorectum, 7 lung, 6 brain and 49 at all other sites combined (with 5 or fewer at any individual site).’